# Annulus Fibrosus Injury Induces Acute Neuroinflammation and Chronic Glial Response in Dorsal Root Ganglion and Spinal Cord—An In Vivo Rat Discogenic Pain Model

**DOI:** 10.3390/ijms25031762

**Published:** 2024-02-01

**Authors:** Alon Lai, Denise Iliff, Kashaf Zaheer, Jennifer Gansau, Damien M. Laudier, Venetia Zachariou, James C. Iatridis

**Affiliations:** 1Leni and Peter W. May Department of Orthopedics, Icahn School of Medicine at Mount Sinai, New York, NY 10029, USA; denise.iliff@icahn.mssm.edu (D.I.); kashaf.zaheer147@gmail.com (K.Z.); jennifer.gansau@mssm.edu (J.G.); damien.laudier@mssm.edu (D.M.L.); james.iatridis@mssm.edu (J.C.I.); 2Department of Pharmacology, Physiology and Biophysics, Chobanian and Avedisian School of Medicine at Boston University, Boston, MA 02118, USA; vzachar@bu.edu

**Keywords:** discogenic pain, intervertebral disc degeneration, annulus fibrosus injury, spinal cord, dorsal root ganglion, animal model, macrophage, astrocyte, microglia, satellite glial cell

## Abstract

Chronic painful intervertebral disc (IVD) degeneration (i.e., discogenic pain) is a major source of global disability needing improved knowledge on multiple-tissue interactions and how they progress in order improve treatment strategies. This study used an in vivo rat annulus fibrosus (AF) injury-driven discogenic pain model to investigate the acute and chronic changes in IVD degeneration and spinal inflammation, as well as sensitization, inflammation, and remodeling in dorsal root ganglion (DRG) and spinal cord (SC) dorsal horn. AF injury induced moderate IVD degeneration with acute and broad spinal inflammation that progressed to DRG to SC changes within days and weeks, respectively. Specifically, AF injury elevated macrophages in the spine (CD68) and DRGs (Iba1) that peaked at 3 days post-injury, and increased microglia (Iba1) in SC that peaked at 2 weeks post-injury. AF injury also triggered glial responses with elevated GFAP in DRGs and SC at least 8 weeks post-injury. Spinal CD68 and SC neuropeptide Substance P both remained elevated at 8 weeks, suggesting that slow and incomplete IVD healing provides a chronic source of inflammation with continued SC sensitization. We conclude that AF injury-driven IVD degeneration induces acute spinal, DRG, and SC inflammatory crosstalk with sustained glial responses in both DRGs and SC, leading to chronic SC sensitization and neural plasticity. The known association of these markers with neuropathic pain suggests that therapeutic strategies for discogenic pain need to target both spinal and nervous systems, with early strategies managing acute inflammatory processes, and late strategies targeting chronic IVD inflammation, SC sensitization, and remodeling.

## 1. Introduction

Chronic low back pain involves discogenic pain, or back pain with intervertebral disc (IVD) degeneration (IVDD) as the main diagnosis, in about 40% of cases [1,2,3,4]. Discogenic pain is a multifactorial and complex condition that is difficult to manage pharmacologically and not well indicated for surgical interventions [5,6]. Structural defects in the annulus fibrosus (AF) and endplate (EP) are specific IVDD phenotypes known to contribute to discogenic pain, and the presence of structural IVD defects and pain distinguishes pathological IVD degeneration from aging [5,6]. AF defects can cause painful processes due to spinal instability (e.g., from nucleus pulposus (NP) depressurization, IVD height loss, or other degenerative changes that cause nerve root compression and irritation) and pro-inflammatory conditions (e.g., from incomplete IVD healing, progressive matrix degradation, neuronal sensitization, and neurovascular ingrowth) [5]. While these causes of painful IVDD have been identified and are broadly known, surprisingly little is known about the timing of those changes or the specific changes to involved spinal tissues. The progression from acute pain to chronic pain is not well-understood and is important for identifying tissue targets and timing windows of these pathological changes in order to inform strategies when therapeutic interventions for acute and chronic discogenic pain are likely to have the most efficacy.

There are no disease modifying treatments for chronic low back pain with discogenic origins, and little consensus about the best therapeutic strategy or timing for intervention due to its complex etiology [7]. Conservative therapies are commonly prescribed as initial approaches for managing acute non-severe back pain, while surgical interventions, such as spinal fusion, discectomy, and total disc replacement, are usually considered as the last resort for discogenic pain only when conservative treatments have proven ineffective, and physical examination can correlate imaging findings directly to disability. Both conservative treatments and surgical interventions can alleviate some aspects of pain progression for herniation and IVDD, but neither of them shows significant improvements of pain as compared to baseline levels prior to treatment [8]. Outcomes of spinal fusion surgery for discogenic pain have highly variable success rates post-operatively, ranging from 35% to 92% in multiple studies [9,10,11,12,13,14]. Such high variability for successful discectomy and fusion surgeries suggests that the IVD is only one source of pain and disability, and only in some cases. Importantly, the ineffectiveness of surgical interventions targeting chronic IVDD, including discectomy and fusion, further highlights that degenerated IVDs are not the sole source of pain [15,16,17].

Chronic back pain involves interactions between the IVD and surrounding neuronal structures. The IVD is innervated with nociceptive fibers connecting from the dorsal root ganglion (DRG) to the spinal cord (SC) dorsal horn which can also be sources of pain. Inflammation, mechanical stress, or structural changes associated with IVD degeneration can activate the nerve endings in the outer AF and endplate that are sensitive to pain and pressure and send nociceptive signals to the DRGs and the SC dorsal horn [5,18,19]. Repeated activation of nociceptive pathways can lead to peripheral (associating with DRG) and central (associating with SC) sensitization, causing a chronic pain condition [20,21]. Nerve sensitization typically involves increased expression of neuropeptides such as Substance P (SubP) and calcitonin gene-related peptide (CGRP) [20]. The increased presence of these neuropeptides in peripheral and central nervous system tissues can amplify the perception of pain and make it more persistent, even after the initial source of pain has been addressed [22,23].

DRG and SC sensitization and neuroinflammation are critical features of neuropathic pain models, but are not well-investigated in discogenic pain models. Neuronal remodeling as well as peripheral and central sensitizations are widely characterized in neuropathic pain models, which focus specifically on understanding the changes in pain processing and perception [24,25,26,27,28]. Neuropathy from spared nerve injury increases neuropeptides, while also increasing the presence of immune cells (macrophages and microglia) and glial cells (satellite glial cells and astrocytes), and all of these cells play roles in amplifying pain signals and contribute to the development and maintenance of chronic pain states [29,30]. Both microglia and astrocytes are responsible for the production of neuron-derived proinflammatory mediators, including tumor necrosis factor alpha (TNFα), interleukin-1 beta (IL-1β), colony-stimulating factor 1 (CSF1), caspase-6, and brain-derived neurotrophic factor (BDNF) [20,31]. Repeated nociceptive input from these mediators following the activation of microglia and astrocytes can decrease the pain threshold within a ‘sensitized’ spinal cord [23]. In studies involving peripheral inflammation and nerve injury models, spinal cord astrocytes (measured with glial fibrillary acidic protein, GFAP, intensity) and microglia (measured with ionized calcium-binding adaptor molecule 1, Iba1, intensity) were found to be activated and elevated, correlating with increased pain-like behaviors indicating sensory hypersensitivity [32,33,34,35]. This suggests that neuroinflammation plays a significant role in both inflammatory and neuropathic pain. While Substance P, microglia, and astrocytes are recognized as important spinal cord cells involved in neuropathic pain pathogenesis, limited knowledge exists regarding their roles in chronic discogenic pain [36,37,38,39,40,41].

Animal model studies have shown that IVDs are innervated by the surrounding peripheral and central nervous systems, indicating that IVDs can be a direct source of discogenic pain, and/or interact with surrounding neuronal structures [42,43]. AF puncture injury is a well-established and widely recognized animal model for studying discogenic pain and IVD degeneration. Studies utilizing AF-injured IVDs exhibited IVDD with IVD height loss, altered structures, and morphological changes, increased pro-inflammatory cytokinesis, and matrix-degrading enzymes, and the severity of these degenerative changes can be impacted by the size of the needle puncture, the puncture depth, needle motion, and/or the number of times the IVD is punctured [44,45,46,47,48,49,50,51,52,53]. IVD injury also showed pain-related behaviors with mechanical allodynia and thermal withdrawal latency [44,46,54]. Moreover, AF puncture injury can enhance sensory hypersensitivity with increased nociceptive neuropeptides in the peripheral nervous system with increased CGRP [55,56,57,58,59,60,61] and SubP in animal lumbar DRGs [44,62]. AF injury models have also shown an increase in ionized calcium binding adaptor molecule 1 (Iba1) and glial fibrillary acidic protein (GFAP) in lumbar SC [60], which are glial cell markers for microglia and astrocytes, respectively, and are highly associated with neuroinflammation and neural remodeling in the central nervous system. GFAP is a key intermediate filament protein present in various types of glial cells, including astrocytes in the central nervous system and satellite glial cells (SGC) in the peripheral nervous system [29]. GFAP makes up the cytoskeletal framework to maintain the structural integrity of glial cells and regulates concentrations of neurotransmitters and ions for optimal neuronal function during normal physiological conditions. In response to nerve injury and inflammatory pain, GFAP demonstrates upregulation in both SGC and astrocytes, indicating activation of both SGC and astrocytes. These activated glial cells release various signaling molecules, including cytokines and chemokines, along with increased production of potassium and glutamate, thereby sensitizing neurons, amplifying their responsiveness and contributing to the manifestation of pain [29]. While animal models of IVDD demonstrate pain-related behaviors, very few studies investigated the changes in the DRG or SC which are associated with sensitization, and the dynamic interactions of all three tissues might contribute to painful conditions [44,45,63,64].

The overall objective of this study was to investigate the changes in the spine, DRG, and spinal cord at from acute (3 days to 2 weeks) and chronic conditions (8 weeks) in vivo using a rat discogenic pain model with surgically-induced AF injury. The spine and IVD morphological changes were evaluated histologically, while the nociceptive, inflammatory, and glial responses in spine, DRG, and spinal cord were assessed immunohistochemically. Spine inflammatory response was assessed via CD68 as a pan-macrophage marker. In both DRG and SC, SubP, Iba1, and GFAP were used as markers for nociception, neuroinflammation, and neuronal remodeling, respectively. Lastly, the time course of changes in these markers across tissues as well as crosstalk between IVD, DRG, and SC were evaluated. We hypothesized that IVD injury will result in acute inflammatory responses in IVD, DRG, and SC that progress to chronic DRG and SC sensitization and remodeling. Investigating these early and late changes in the IVD, DRG, and SC can help elucidate timing windows and target tissues for improved therapies for discogenic pain.

## 2. Results

### 2.1. AF Puncture Injury Did Not Affect Rat General Health

The results of the AF injury group at different time points were compared with the naïve control group which did not receive any surgical interventions. After surgery, the animals were closely monitored for any indications of distress or changes in behavior that might signify pain or discomfort, including mobility, staggering, and interaction with the researcher. Notably, these observations did not reveal any discernible differences when compared with the naïve rats. Moreover, there was no significant change in body conditioning score throughout the experiment, with no observable intraoperative complications. Body weight was not obtained in this study to avoid additional handling and stress for the animals, since our previous studies with this injury model [44,45,46,47], consistently demonstrated a continuous increase in body weight with no significant differences between groups at any time points post-injury. Together, these findings strongly suggested that the procedures of AF injury were well tolerated by the rats and did not significantly affect their general health.

### 2.2. AF Puncture Injury Induced IVD Degeneration

The rat lumbar spines were isolated and stained with safranin-O/fast-green/hematoxylin to determine the progression of IVD morphological changes following AF injury by comparing to those from the naïve group. The IVDs from naïve rats or uninjured IVD levels (i.e., L2–3 IVD) from rats in the AF injury group exhibited low IVD degeneration scores and had normal morphology, including clear NP-AF boundary, glycosaminoglycan (GAG)-rich NP with notochordal cells, and well-organized AF lamella with fibroblastic annular cells (Figure 1A). Injury significantly increased IVD degeneration score by 3 days and caused moderate to severe IVD degenerative changes including a smaller and more fibrotic NP with fewer and/or clustered NP cells, a less distinct NP-AF boundary, and a disorganized AF with ruptured lamellae along the puncture track (Figure 1A). The puncture track also exhibited decreased GAG content and fewer annular cells (Figure 1A). EP disruption and defects were observed in some injured IVDs. No bulging or herniation was observed histologically, although GAG-rich tissue was observed outside the anterior outer AF and/or within anterior longitudinal ligament in 2 out of the 3 injured IVDs at 3 days post injury; however, this was not observed in the injured IVDs isolated at 1, 2, or 8 weeks post injury, suggesting that any GAG released from the severe AF puncture injuries was rapidly resorbed (Figure 1A).

The severity of IVD degeneration was quantified using a semi-quantitative degeneration grading system specifically for rat IVDs. The degeneration grading system evaluated NP morphology, NP cellularity, NP-AF border, AF morphology, and EP irregularity, using a score between 0 and 2 for each sub-category, with 0 representing normal morphology and 2 representing severe degenerative characteristics [65]. The quantitative analysis showed that, compared to the IVDs from the naïve group, degeneration scores were significantly higher in the AF-injured IVDs at all time points (Figure 1B, Appendix A). The IVD histomorphology showed that IVD degeneration was a progressive and cumulative change from 3 days to 8 weeks (Figure 1A); however, the quantitative degeneration score detected an immediate and significant increase at 3 days that did not progress and remained constantly elevated through 8 weeks (Figure 1B), suggesting this degeneration grading system did not have the granularity to detect the finer progressive changes over time.

### 2.3. AF Puncture Injury Induced Macrophage-Associated Inflammatory Responses in the Spine

CD68 is a marker for macrophages and monocytes, and an increase in the level of CD68-immunoreactivity (-ir) indicates increased macrophage-associated inflammatory response. The CD68-ir was quantified using a semi-quantitative grading scale modified from our previous study [66] that utilizes four sub-categories according to the spine regions of AF, endplate, longitudinal ligament, and vertebra, and ranged from 0 to 4, with 0 representing no staining and 4 representing extensive staining (Appendix A). The spines from naïve rats exhibited a few CD68 immunopositive cells in the AF, endplate, and longitudinal ligaments, while there were abundant cells in the vertebral body that showed CD68-ir (Figure 2A). Compared to the naïve spine, there was relatively more CD68-ir in the spines following AF injury, mainly in the regions of the anterior AF, anterior longitudinal ligament, and endplate (Figure 2A). The highlighted images further emphasize these CD68 positive vs. negative regions within the anterior AF, anterior longitudinal ligament, and endplate (Figure 2A). These changes were significant in the anterior AF from 3 days to 8 weeks. However, the CD68-ir were peaked at 3 days and 1 week and returned to naïve levels by 8 weeks in the endplate and anterior longitudinal ligament, respectively (Figure 2B). Slightly more CD68-ir was also observed in the posterior AF and posterior longitudinal ligament (Appendix A); however, there were very few or almost no CD68 immunopositive cells in the NP. The significant increase in endplate CD68-ir indicates involvement in this AF injury model, and the notable CD68-ir peak at 3 days suggests a potential role for macrophage from the endplate into the IVD. The results reveal that the AF injury induced time-dependent macrophage-associated inflammatory response within the spine, particularly in the injured anterior AF and anterior longitudinal ligament regions.

### 2.4. AF Injury Induced Temporary Increase in SubP in DRG

SubP is an important nociceptive neuropeptide in both peripheral and central nervous system associated with pain perception and processing. In the DRG, SubP-ir was mainly observed in the small neurons (Figure 3A) which are nociceptive cells, via small unmyelinated C-fibers or thinly myelinated Aβ fibers and play a significant role in the transmission of pain signals from peripheral tissues to the nervous system. AF injury significantly increased the percentage of SubP-ir in L2 DRG, which peaked at 3 days after injury (Figure 3B), then significantly decreased with no significant difference from the naïve group at 1 week, 2 weeks, or 8 weeks (Figure 3B). The rapid and acute response of DRG SubP at 3 days suggests it might play a role in the initial acute response following AF injury and the development of acute pain states.

### 2.5. AF Injury Induced Temporary Neuroinflammatory Response in DRG

Elevated Iba1-ir in DRG indicate an increased number of activated macrophages and neuroinflammatory response. Iba1-ir was mainly distributed towards the middle of DRG, outside of the DRG neurons (Figure 4A). The percentage of Iba1-ir in the DRG significantly increased and peaked at 3 days after injury (Figure 4B), then significantly decreased to levels similar to those observed in the Naïve group at 1 week, 2 weeks, or 8 weeks (Figure 4B). The changes of Iba1-ir in the DRG indicate a dynamic time-dependent response in the activation of macrophages and an acute neuroinflammatory response.

### 2.6. AF Injury Induced Sustained Increase in Satellite Glial Cell in DRG

GFAP is a specific marker for activated satellite glial cell (SGC) in the DRG. Elevated GFAP expression may be interpreted as a measure of neuronal remodeling associating with neuroinflammatory processes and peripheral sensitization. In the DRG, GFAP-ir was mainly observed as a cellular sheath surrounding the large neurons which mainly play a role in non-nociceptive signals via thickly myelinated Aβ fibers (Figure 5A). The percentage of GFAP-ir significantly increased at 3 days following AF injury and remained at this elevated level until the 8 week time point (Figure 5B). The changes of GFAP-ir indicate a persistent response of activated SGCs in response to AF injury, which might be associated with the development and persistence of pain.

### 2.7. AF Injury Induced Persistent Spinal Cord Sensitization

In the spinal cord, SubP-ir was localized mainly in lamellae I and II of the spinal dorsal horn (Figure 6A), which are the terminations of nociceptive A-delta and C fibers. The highlighted images (Figure 6A) show increased intensity and density of SubP in spinal dorsal horn lamellae I and II from 2 weeks to 8 weeks compared to naïve rats. Quantitatively, AF injury significantly increased the SubP-ir in the spinal dorsal horn gradually from 3 days to 2 weeks following AF injury and remained elevated until the 8 week time point (Figure 6B). The changes in SubP in response to AF injury in this region indicate an activation of nociceptive A-delta and C fibers, suggestive of central sensitization.

### 2.8. AF Injury Induced Temporary Microglia-Mediated Neuroinflammatory Response

Iba1 is a marker for activated spinal cord microglia. An increased level of Iba1-ir indicates an increased number of activated microglia, indicating a neuroinflammatory response. Iba1-ir was evenly distributed in the spinal cord dorsal horn (Figure 7A). The highlighted Iba1 positive images (Figure 7A) show increased Iba1 surrounding neurons in the spinal dorsal horn, particularly at 2 weeks following injury compared to all other time points. AF injury revealed a wave of microglia-mediated neuroinflammatory responses through a gradual increase in the percentage of Iba1-ir in the spinal dorsal horn from 3 days to 2 weeks which significantly decreased by 8 weeks, returning to naïve levels (Figure 7B). The changes in Iba1-ir in the spinal dorsal horn show dynamic and time-dependent microglial activation and neuroinflammatory response.

### 2.9. AF Injury Induced Sustained and Progressive Astroglial Response

GFAP is a marker for activated spinal cord astrocytes, which plays a significant role in neuroinflammatory processes, central sensitization, and chronic pain conditions. Similar to Iba1, GFAP-ir was also evenly distributed in the spinal cord dorsal horn (Figure 8A). The highlighted images (Figure 8A) show the typical star-shaped morphology of astrocytes surrounding the neurons as well as an increased SC GFAP in the dorsal horn from naïve to 8 weeks following injury. The percentage of GFAP-ir gradually and continuously increased from 3 days to 8 weeks after AF injury and became statistically significant at 1 week, 2 weeks, and 8 weeks compared to the naïve group (*p* < 0.05, *p* < 0.01, and *p* < 0.0001, respectively) (Figure 8B). The significant and sustained increase in GFAP-ir indicates that astrocyte activation is a persistent response to AF injury.

### 2.10. Crosstalk between Spine, DRG, and SC

To determine the relationship of inflammatory cells and sensitization between spine, DRG, and SC, the temporal patterns of changes in spine CD68-ir were compared to the changes in SubP-, Iba1-, and GFAP-ir in the DRG and SC. Compared to the naïve group, the increase in CD68-ir in spine anterior AF at 3 days post-injury was similar to the change in Iba1 in the DRG, indicating that inflammatory cells were present in the spine and DRG simultaneously, and suggesting a likelihood of inflammatory crosstalk between the spine and DRG during the early phase of injury (Figure 9). On the other hand, the changes in Iba1-ir were delayed in the SC and peaked at 2 weeks post-injury, suggesting this inflammatory crosstalk from IVD and DRG to SC took more time due to the greater anatomical distance between these tissues. Together, we identified an immediate and synchronized response between the spine and DRG and a delayed central SC response.

The changes in DRG and SC were distinct between acute and chronic states. At the acute state, the DRG SubP-, Iba1-, and GFAP-ir were significantly increased along with spine CD68-ir (Figure 10A), suggesting AF injury causes spine inflammation that results in substantial and rapid changes to DRG sensitization, neuroinflammation, and remodeling.

In the chronic state, the spine CD68-ir, DRG GFAP-ir, as well as SC SubP- and GFAP-ir were significantly increased from AF injury compared to the naïve group (Figure 10B). These results suggest that the AF injury causes sustained increases in spine inflammation resulting in sustained and chronic DRG remodeling, and SC sensitization and remodeling. Together, AF injury results in rapid and acute DRG changes and chronic and sustained central SC involvement, and these peripheral and central nervous system changes are associated with spinal inflammation that rapidly occurs and does not resolve in these degenerating IVDs.

## 3. Discussion

Chronic discogenic pain is a common musculoskeletal disorder that is refractory to surgical and therapeutic strategies focusing solely on the IVD, highlighting the need look beyond the spine to understand which tissues are involved and how they interact to produce pain. This is the first study to comprehensively investigate the time course of sensitization, inflammation, and remodeling processes in the spine and peripheral and central nervous systems in an AF injury-induced discogenic pain model. Changes in sensitization and inflammation were time-dependent and tissue-dependent. AF injury caused acute spinal inflammation at the injury site and EP, and chronic lack of healing, progressive degeneration, and sustained inflammation. The peripheral nervous system was involved acutely with rapid and transient DRG sensitization, inflammation, and remodeling. The central nervous system was involved chronically, and in SC sensitization and remodeling, and these SC changes were likely involved with chronically increased spine inflammation and DRG remodeling. These findings suggest distinct early and late changes to IVD and neuronal tissues in discogenic pain conditions, highlighting why this condition is so complex and often refractory to treatments focused on a single organ system. The changes observed in IVD, DRG and SC suggest potential strategies and targets required for managing acute and chronic discogenic pain.

AF injuries in rats and mice are known to cause rapid and sustained behavioral changes suggesting discogenic pain [45,63,67,68,69]. This study advanced this knowledge to show that AF injury triggered an immediate and broad inflammatory reaction that was likely a contributor to this known painful response. Specifically, AF injury caused nearly immediate spine inflammation (CD68-ir), and simultaneously increased DRG inflammation (Iba1-ir) and subsequent SC inflammation (Iba1-ir). The increased macrophages increased broadly in the affected tissues and also involved DRG SubP and GFAP (in SGC), suggesting peripheral sensitization that could be a contributor to pain-like behaviors. Similar studies utilizing an IVD injury model of rat L5–6 IVD puncture showed increased pro-inflammatory cytokines of tumor necrosis factor alpha (TNFα) and interleukin-6 at 1 and 4 days post-injury, along with sustained increased DRG CGRP [69]. Chronically, AF injury caused progressive IVD degeneration and chronic spinal inflammation with central sensitization involving increased SC SubP and GFAP (astrocytes) along with chronically elevated DRG GFAP (SGCs) sustained at least until 8 weeks post-injury. Prior studies with lumbar IVD puncture injuries in both rats and mice also showed central sensitization with increased SC microglia (Iba1) at early and late time-points post-injury, and these changes were associated with impaired grip force and physical function [70]. Together with the literature, our results indicate that IVD injury involves acute macrophage recruitment, pro-inflammatory cytokines, and acute pain with DRG sensitization and remodeling. In the chronic phase, the injured IVD involves persistent presence of spine macrophages, along with DRG and SC remodeling and SC sensitization, which might lead to chronic pain.

The IVD, DRG, and SC changes observed in this study also occurred in a spontaneous model of IVDD, suggesting that these AF injury-induced changes are likely to be important in discogenic pain more broadly and to involve inflammatory cell cross-talk between these spinal tissues [69]. Specifically, an aged SPARC-null mouse model exhibited IVD degeneration and increased pain-related behaviors with hypersensitivity to cold, axial discomfort, and motor impairment, along with increased CGRP and neuropeptide-Y in the DRG, and central sensitization with increased SC CGRP, microglia (Iba1), and astrocytes (GFAP). Taken together, discogenic pain is likely to involve localized spinal inflammation from IVDD and inferior IVD healing, DRG sensitization, neuroinflammation, and remodeling, and chronic SC sensitization and remodeling. Furthermore, the DRG changes are likely implicated in acute pain, while the SC changes are implicated in chronic pain. Moreover, both acute and chronic responses could be driven at least in part by the spinal inflammation since this study highlights the increased inflammatory cells in the spine, DRG, and SC. The spine macrophages were mainly increased in the longitudinal ligament, AF, and endplate, which are known to be highly innervated with nociceptive A-delta and C fibers extending from the DRG [71] and extending into the SC. We speculate that the nerve fibers might be a direct pathway for the spine-DRG-SC crosstalk, and the sustained spine inflammation (at anterior AF) might be associated with the sensitizations in DRG and SC.

AF injury and IVD degeneration may induce neuropathic pain-like conditions in the peripheral and central nervous systems. Sensitization and neuroinflammation in the DRG and SC are well-documented for neuropathic pain. A neuropathic pain model involving partial sciatic nerve ligation in mice exhibited increased pain-related behaviors of mechanical allodynia, which were accompanied with acute increases in DRG macrophages (Iba1), followed by delayed increases in SC microglia (Iba1), and chronic sustained activation of astrocytes (GFAP) in the SC [72]. The spatial and temporal changes in SC microglia and astrocytes in this study parallel those seen in peripheral nerve injury models of sciatic nerve constriction [35], sciatic nerve transection [73], and L5 nerve transection [74], as well as a diabetic neuropathic pain model [75]. The similarities in the responses of early microglial activation and sustained astrocyte activation in the SC between discogenic pain and neuropathic pain models reveals that IVDD appears to have neuropathic components and should be viewed not only as a pathological condition affecting the spine, but also as a condition that significantly impacts the peripheral and central nervous system tissues. SGCs and astrocytes have distinct roles in the peripheral and central nervous systems, but both are shown to become activated in response to neuronal pathologies, likely contributing to the modulation of neuronal and synaptic sensitivity and increased production of pro-inflammatory cytokines following injury. Therefore, sustained activation of SC astrocytes and DRG SGCs can induce central and peripheral sensitizations, contributing to the persistence of chronic pain [29]. Moreover, these similarities also provide insight into why current treatments for discogenic pain, which primarily address the spine without modulating chronic remodeling of the nervous system, may not fully alleviate chronic pain.

The distinct responses observed in spine, DRG, and SC at early and late timepoints following IVD injury reveal that therapeutic interventions are likely to be most effective if they manage all affected tissues and consider the timing of intervention. The immediate proinflammatory reaction in the spine and DRG followed by a later reaction in the SC in this and other studies [44,69,70], points toward the potential of early anti-inflammatory interventions being useful to treat discogenic pain. Localized administration of the TNFα inhibitor, Infliximab, at the time of AF puncture injury in a rat model mitigated IVD degeneration and mechanical allodynia [46], suggesting that early treatments altering the early inflammatory responses offer the potential to inhibit chronic discogenic pain. Intraperitoneal administration of Infliximab at the time of injury in a rat IVD herniation model also demonstrated alleviation of pain-related behaviors and brain-derived neurotrophic factor, a neurotrophin involved in nociceptive signaling, in both the DRG and SC at an acute time point [76,77] further suggesting early anti-inflammatories may be beneficial. Etanercept, another TNFα inhibitor, applied directly into injured rat IVDs also alleviated DRG CGRP [57]. Therefore, this study suggests spinal and neural inflammation from AF injury is involved with pain, and the preclinical literature suggests therapeutic interventions aimed at modulating inflammatory reactions during acute injury hold promise for managing acute injury and preventing the transition to chronic pain.

Human clinical studies on anti-inflammatory treatments for discogenic pain and radiculopathy are more mixed or inconclusive [78,79,80], suggesting a need to consider the timing and type of treatment. Furthermore, a recent study suggests that broad-acting steroidal and non-steroidal anti-inflammatory interventions may lead to prolonged pain states, with neutrophils playing a role in protecting against the development of chronic pain [81]. Consequently, future research is needed to explore a more limited course of anti-inflammatories, or a more specific anti-inflammatory intervention, that can better balance the competing needs of promoting spinal repair while inhibiting long-term neuronal sensitization and remodeling. Furthermore, this discogenic pain model and preclinical neuropathic pain models show that sensitization and SGC activation in the DRG were immediate, within 2–48 h, after injury [82], while SC astrogliosis occurs more gradually at a later time point following microglial activation and the release of inflammatory mediators [35,83], suggesting that the time window for early intervention to alleviate DRG sensitization is short. Furthermore, it is possible to indirectly demonstrate the involvement of peripheral and central nervous systems for discogenic pain in humans. SC and DRG stimulations are neuromodulation techniques and an established therapy for treating neuropathic pain. Electrodes in the epidural space stimulate targeted zones of the spinal cord dorsal columns or DRGs to produce paresthesia, and pain relief has been thought to result from overlapping paresthesia with the area of pain [84]. This technique has been recently adopted and reported to be effective for improving pain, functional status, quality of life, and medication consumption in patients with chronic low back pain [84,85,86,87]. Therefore, a treatment strategy could include early interventions for acute back pain with limited duration anti-inflammatories to inhibit spinal inflammation and DRG neuroinflammation and later neuropathic interventions focused on preventing or modulating the activation of SC astrocytes and transition to chronic discogenic pain. Such a treatment strategy needs further preclinical research on feasibility of treatment strategies and requires innovations in clinical thinking to identify and treat patients more rapidly following acute back pain or spinal injury episodes.

Important parallels exist between this preclinical study and the human conditions, even though rat and human studies cannot be directly compared. The relevance of this study to the human clinical condition is to highlight the complex and interacting involvement between spinal tissues and DRG and SC, the likelihood that discogenic pain involves substantial DRG and SC neuropathy, and the need to consider earlier interventions to acute back pain episodes. The surgically induced AF injury resulted in IVD degenerative changes and changes to the surrounding endplates and ligaments with many similarities to the spontaneous SPARC-null mice [60], suggesting likely relevance to the human IVD degeneration phenotypes that accumulate more slowly. It is difficult to directly compare the changes in DRG and SC from this AF injury-driven IVD degeneration model to the human condition, since, to our knowledge, no study has been conducted to determine the changes in the DRG and SC in patients with acute and chronic discogenic pain. The human nervous system is intricate and difficult to study directly, and ethical considerations limit invasive experiments on humans. This highlights the importance of animal models to provide a controlled and ethically acceptable way for neuroscience studies. However, the preclinical results on DRG and SC changes in this model provide likely sources of chronic pain. The likelihood of DRG and SC involvement in chronic human discogenic back pain provides and explanation why outcomes of spinal fusion surgery for discogenic pain have highly variable success rates, since spinal fusion surgery removes the degenerated IVD but does not address the glial remodeling and sensitization of DRG and SC that are likely to be involved in persistent pain in patients with poor outcomes. We believe preclinical models like this are best used to understand spinal pathophysiology, and then to inform and screen treatments for eventual use to improve discogenic pain care treatments in humans.

Some limitations of the current study warrant discussion. This time course study utilized a naïve group as a control since it was unclear which time point and which effects might be prominent. Therefore, we used naïve rats as a baseline control that did not receive any surgical interventions. Results indicated that this study was sufficiently controlled since DRG SubP, DRG Iba1, SC Iba1, and many of the spinal CD68 levels recovered to naïve levels by 8 weeks. While we cannot fully reject the effects of abdominal incision on SC and DRG SubP, we note that our prior studies using this discogenic pain model identified that sham surgery (exposing lumbar IVDs without any IVD injury) did not induce significant IVD degeneration, IVD interleukin-1b content, hindpaw withdrawal threshold, or CGRP expression in L2 DRG at both 1 and 6 weeks post-surgery [47,88], suggesting that the significant changes observed in this study are attributable to the AF-injury-induced IVDD model. Additionally, we did not measure pain-related behaviors in this study because this model has been previously well-characterized for increased mechanical and thermal allodynia over time [44,45,46,47], and it was our priority to evaluate immunohistochemistry on DRG and SC as biochemical markers of pain in this study. The SC changes in this study were very similar to those observed in a rat EP microfracture injury (at 8 weeks) [35,74], and a rat neuropathic pain study with L5 nerve transection [74], and both of these models demonstrated enhanced pain-related behaviors including mechanical hindpaw allodynia and reduced axial grip force. Having identified chemical markers of pain in this study, we foresee future interventional studies with this to include pain-related behavior assays and chemical markers of pain, sensitization, and neuroinflammation as output measurements. Furthermore, this study used male rats to reduce the variance in observed effects, although future studies will need a female cohort to discern the generalizability of the findings across sexes, since some sex-based differences in the mechanisms of pain, neuroinflammatory processes, and healing responses between males and females have been reported [47,62,89,90].

## 4. Materials and Methods

### 4.1. Study Design

All experimental procedures were approved by the Institutional Animal Care and Use Committee. Skeletally mature male Sprague–Dawley rats (n = 37; 5–6 months old) were randomly assigned into naïve or AF injury groups (Figure 11). The rats in the AF injury group underwent annular puncture surgery, and they were euthanized at 3 days (n = 8), 1 week (n = 8), 2 weeks (n = 5), and 8 weeks (n = 8) post-injury, in order to better understand the crosstalk between the spine, DRGs, and SC, this study investigates the time course of changes in the progression from acute IVD injury to chronic painful conditions. The rats in the naïve group (n = 8) did not receive any surgical interventions, and they were used as control group to determine the effects of IVD injury on the rats. After euthanasia, the lumbar spines (L2–L6), L2 DRGs, and lumbar SC (corresponding to vertebral level T12-L1) were dissected for histology to evaluate the severity of IVD degeneration, as well as immunohistochemical analyses for inflammatory responses and neuronal sensitization.

### 4.2. Surgical Procedure and AF Puncture Injury

Surgical procedures were performed under aseptic conditions and general anesthesia via 2% isoflurane (Baxter, Deerfield, IL) [44]. The anterior aspect of the lumbar spine was exposed using an anterior abdominal incision through the skin and peritoneal membrane. The L3–6 IVDs were identified preliminarily using a pre-operative anterior-posterior X-ray image, and then confirmed using anatomical landmarks of aortic trifurcation and iliac crest. The L3–4, L4–5, and L5–6 IVDs were each punctured three times (i.e., midline anteriorly as well as left and right anterolaterally) using a 26 G needle at a depth of 3 mm guided by a needle stopper including needle twisting and side-to-side motion, followed by an intradiscal injection of sterile PBS to induce more severe tissue disruption and degenerative changes [62] using a calibrated microliter syringe (Hamilton Company, Reno, NV, USA) (Figure 11). The abdominal muscles were closed using 3-0 silk sutures and the skin was closed using 4-0 nylon sutures. The rats were allowed ad libitum access to food and water and were closely monitored for complications. All animals were allowed unrestricted movement in their cages for the entire experimental duration and housed two per cage with the exception of the 24 h post-operative period, when animals were singly housed [62].

### 4.3. Tissue Collection

At the end of the experiment, all rats were transcardially perfused with acetic zinc formalin (Newcomer Supply, Middleton, WI, USA) while the animals were under general anesthesia. The lumbar spine, DRGs, and spinal cord were dissected and fixed in acetic zinc formalin for at least 48 h. The formalin-fixed lumbar spines were used for spine morphology and IVD degeneration grading, and immunohistochemistry for CD68, a pan-macrophage marker, to evaluate the inflammatory responses associated with macrophage/monocytes. The L2 DRGs and lumbar SC were used for evaluating the presences of substance P (SubP, a neuropeptide marker of nociception), ionized calcium binding adaptor molecule 1 (Iba1, a marker for macrophages/microglia), and glial fibrillary acidic protein (GFAP, a marker for satellite glial cells/astrocyte) using immunohistochemistry.

### 4.4. Spine Morphology and IVD Degeneration Grading

The formalin-fixed spine specimens were decalcified in formic acid over 3 days with 3 changes, embedded in paraffin, and sectioned sagittally at 5 μm intervals. The mid-sagittal sections with the AF puncture injury were identified and stained with safranin-O/fast-green/hematoxylin for visualizing GAG content, IVD morphology, and IVD cellularity. The stained slides were imaged using bright-field microscopy (Leica Microsystems, Inc., Deerfield, IL, USA). The IVD degeneration score was determined using a semi-quantitative grading system to evaluate NP morphology, NP cellularity, NP-AF border, AF morphology, and EP irregularity [65]. IVD degeneration grading was performed by an experienced spine researcher who was blinded to the experimental groups. The degeneration scores from the three injured IVDs within the same animal were averaged, and then compared with those isolated from naïve and other AF injury groups.

### 4.5. Immunohistochemical Analysis for Spine Macrophage

The mid-sagittal sections with AF puncture injury were identified, followed by deparaffinization and rehydration. After antigen retrieval with IHC Enzo (ADI950-280-0015, ENZO Life Science, Farmingdale, NY, USA) and protein block (X0909, Agilent Technologies, Inc., Santa Clara, CA, USA), the sections were incubated for 1 h at room temperature with rabbit monoclonal primary antibody against rat CD68 (1:500 dilution, ab283654, Abcam, Waltham, MA, USA), or normal rabbit serum (S-500-20, Vector Laboratories, Inc., Burlingame, CA, USA) as the negative control. After incubation with horseradish peroxidase-conjugated anti-rabbit secondary antibody (MP-7061, Vector Laboratories, Inc., Burlingame, CA, USA), the sections were treated with diaminobenzidine-based peroxidase substrate (MP-7601, Vector Laboratories, Inc., Burlingame, CA, USA) to visualize the immunoreactivity. The sections were then counterstained with toluidine blue for spine morphology, dehydrated, mounted, and evaluated using a bright-field light microscope (Axio Imager Z1 microscope, Zeiss, Thornwood, NY, USA).

The CD68-immunoreactivity (-ir) in the rat spine was assessed using a semi-quantitative grading scale which was modified from our previous study [66]. Briefly, the grading scale was between 0 and 4, with grade 0 representing no CD68-ir, and grade 4 representing intense staining of CD68-ir (Appendix A). The CD68-ir in different spine regions of the NP, AF (anterior and posterior), EP (upper and lower), vertebral body (upper and lower), as well as longitudinal ligaments (anterior and posterior) at the levels of L3–4, 4–5, and L5–6 (i.e., the IVD injured spine levels) were evaluated by researchers who were blinded to the experimental groups. The grading of CD68-ir for different spine regions from the three spine levels were averaged for statistical analysis.

### 4.6. Immunohistochemical Analysis for DRG Sensitization and Neuroinflammation

Bilateral L2 DRGs were isolated and used for biochemical analysis of hypersensitization and neuroinflammation since a previous retrograde tracing study demonstrated the DRGs at higher spinal levels innervating from the IVDs at lower levels via paravertebral sympathetic trunks. Therefore, it is believed that the nociceptive and inflammatory responses from the injured L3–4, L4–5, and L5–6 IVDs would transmit and influence L2 DRGs. Both left and right L2 DRGs from each animal were embedded in paraffin, longitudinally sectioned at 5 μm intervals, and used for IHC for different markers.

For SubP, after deparaffinization and rehydration, the DRG sections were treated with an antigen-retrieval buffer of IHC Enzo (ADI950-280-0015, ENZO Life Science, Farmingdale, NY, USA) followed by a protein-blocking buffer of 2.5% normal horse serum (Vector Laboratories, Inc., Burlingame, CA, USA). The sections were then incubated at room temperature for 1 h with mouse monoclonal primary antibody against rat SubP (1:100 dilution, ab14184, Abcam, Cambridge, MA, USA), or negative control with normal mouse serum (NC494H, Biocare Medical, LLC, Pacheco, CA, USA). After removal of primary antibody and buffer washing, the sections were incubated with secondary amplifier antibody with VectaFluor Excel Amplified anti-mouse IgG, Dylight 488 antibody kit (DK-2488, Vector Laboratories, Inc., Burlingame, CA, USA) followed by Dylight488 VectaFluor reagent to visualize the immunoreactivity. The sections were then incubated with NeuroTrace 530/615 red fluorescent Nissl stain (1:200 dilution, N21482, Thermo Fisher Scientific, Waltham, MA, USA) to visualize the neurons and glial cells, being most intense in the nucleoli and rough endoplasmic reticulum of neurons [91]. Nissl staining provides information about the overall distribution and density of neurons in a tissue section and allows us to visualize the morphology of the DRG for analysis. The sections were then mounted using a ProLongTM Gold Antifade Mount with DAPI (P36931, Thermo Fisher Scientific, Waltham, MA, USA).

For Iba1 and GFAP, after deparaffinization and rehydration, the DRG sections were treated with an antigen-retrieval IHC buffer (ADI950-280-0015, ENZO Life Science, Farmingdale, NY, USA) followed by a protein-blocking buffer of 2.5% normal horse serum (Vector Laboratories, Inc., Burlingame, CA, USA). The sections were then incubated at room temperature with rabbit monoclonal antibody against rat Iba1 (1:100 dilution, ab178846, Abcam, Cambridge, MA, USA) for 1 h, rabbit polyclonal primary antibody against rat GFAP (1:300 dilution, ab7260, Abcam, Cambridge, MA, USA) overnight, or negative control with normal rabbit serum (NC495H, Biocare Medical, LLC, Pacheco, CA, USA). After removal of primary antibody and buffer washing, the sections were incubated with secondary antibody of Cy™5 AffiniPure donkey anti-rabbit antibody (1:400 dilution, 711-175-152, Jackson ImmunoResearch Inc., West Grove, PA, USA), and then mounted using a ProLongTM Gold Antifade Mount with DAPI (P36931, Thermo Fisher Scientific, Waltham, MA, USA). All stained slides were imaged using a Leica DM6 B microscope (Leica Microsystems, Inc., Deerfield, IL, USA) at 20× magnification with standardized microscope settings, and all images were analyzed using ImageJ (ImageJ, U. S. National Institutes of Health, Bethesda, MD, USA).

The quantification of individual staining in the DRG involved a precise selection process based on orientation and section level. DRG samples towards the center were chosen for enhanced neuronal density, and axial/ventral sections for equal soma and axon distribution [92]. Exclusion criteria were applied to eliminate sections with primary neuronal areas cut off or containing autofluorescence. Nissl and DAPI counterstains, intensified for better visualization, aided in outlining the region of interest (ROI) containing all neuronal bodies. Using a standardized immunoreactivity (ir) threshold, positive pixels for SubP-ir, Iba1-ir, and GFAP-ir were calculated relative to the ROI area. The analysis incorporated both right and left DRGs, with averaged percentage areas per section. These data were then used to represent values per animal per time point. The sample sizes slightly varied due to excluded samples. For SubP: naïve (n = 5), 3 days (n = 6), 1 week (n = 7), 2 weeks (n = 5), 8 weeks (n = 7). For Iba1: naïve (n = 5), 3 days (n = 7), 1 week (n = 6), 2 weeks (n = 5), 8 weeks (n = 7). For GFAP: naïve (n = 5), 3 days (n = 7), 1 week (n = 7), 2 weeks (n = 5), 8 weeks (n = 7).

### 4.7. Immunohistochemical Analysis for Spinal Cord Sensitization and Neuroinflammation

All fixed spinal cord samples were embedded in paraffin and coronally sectioned at 5 μm intervals. Two sections spread across the lumbar spinal cord were selected from each animal for each antibody.

After deparaffinization and rehydration, each spinal cord section was treated with an antigen-retrieval IHC buffer (ADI950-280-0015, ENZO Life Science, Farmingdale, NY, USA) followed by a protein-blocking buffer of 2.5% normal horse serum (Vector Laboratories, Inc., Burlingame, CA, USA). The sections were then incubated at room temperature for 1 h with mouse monoclonal primary antibody against rat SubP (1:1000 dilution, ab14184, Abcam, Cambridge, MA, USA), rabbit monoclonal antibody against rat Iba1 (1:1000 dilution, ab178846, Abcam, Cambridge, MA, USA), rabbit polyclonal primary antibody against rat GFAP (1:2000 dilution, ab7260, Abcam, Cambridge, MA, USA), or negative control with normal mouse serum (NC494H, Biocare Medical, LLC, Pacheco, CA, USA) or normal rabbit serum (NC495H, Biocare Medical, LLC, Pacheco, CA, USA). After removal of primary antibody and buffer washing, the sections were incubated with secondary amplifier antibody. A VectaFluor Excel Amplified anti-mouse IgG Dylight 488 antibody kit (DK-2488, Vector Laboratories, Inc., Burlingame, CA, USA) was used for SubP, and a VectaFluor Excel Amplified anti-rabbit IgG, Dylight 488 antibody kit (DK-1488, Vector Laboratories, Inc., Burlingame, CA, USA) was used for Iba1 and GFAP. The samples were then incubated with Dylight488 VectaFluor reagent to visualize the immunoreactivity, followed by NeuroTrace 530/615 red fluorescent Nissl stain (1:200 dilution, N21482, Thermo Fisher Scientific, Waltham, MA, USA) to visualize the morphology of the spinal cord to define spinal cord dorsal horns for the analysis. All steps were performed at room temperature and in a dark, moist slide holder. The stained slides were finally mounted using a ProLongTM Gold Antifade Mount with DAPI (P36931, Thermo Fisher Scientific, Waltham, MA, USA) to visualize nuclei. All stained slides were imaged using a Leica DM6 B microscope (Leica Microsystems, Inc., Deerfield, IL, USA) at 20× magnification with standardized microscope settings.

All image analyses were conducted using ImageJ (ImageJ, U. S. National Institutes of Health, Bethesda, MD, USA). For individual staining, two sections with two ROIs for each section (left and right dorsal horn) were examined, analyzed, and averaged for each animal for each staining. Specifically, the left and right spinal cord dorsal horns (lamina I–V, primarily for sensory input reception) for each section were outlined based on guidance from the Nissl stain TXR channel and treated as separate ROIs. Each spinal cord ROI was analyzed using a standardized immunoreactivity (ir) threshold to calculate positive pixels of SubP-ir, Iba1-ir, and GFAP-ir relative to the ROI area. The percentage areas of immunopositive staining from the right and left dorsal horn ROIs were averaged per section, and the averaged values from the two sections were then used to represent a single value per animal per time point for statistical analysis. The samples for spinal cord analysis comprised naïve (n = 8), 3 days (n = 8), 1 week (n = 8), 2 week (n = 5), and 8 week (n = 8).

### 4.8. Statistical Analysis

IVD degeneration score, CD68-ir, as well as SubP-, Iba1-, and GFAP-ir in both DRG and SC were analyzed using one-way ANOVA with Tukey’s multiple comparison test. All statistical analyses were performed using Prism10 (GraphPad, La Jolla, CA, USA), with significance level set at α = 0.05. For crosstalk analysis, the values for each parameter at each timepoint were averaged within each tissue (e.g., spine, DRG, or SC) and then normalized to their respective peak values. The temporal patterns of changes were compared between different tissues.

## 5. Conclusions

This study measured time-dependent structural and molecular changes in the IVD, DRG, and SC following an AF injury to demonstrate substantial interactions between multiple spinal tissues in discogenic pain. Results show that local AF puncture injury induced acute inflammatory crosstalk between the IVD, DRG, and SC and chronic glial response in DRG and SC. The occurrence of acute inflammatory responses suggests a potential early time window for managing IVD and DRG inflammatory changes before central SC changes accumulate.

Furthermore, AF injury also caused chronic persistence of CD68 macrophages in IVDs, likely due to inadequate healing, suggesting a potential mechanism for the sustained increased GFAP in both DRG and SC. This chronic persistence may contribute to the perpetuation of pain states. The central sensitization, characterized by increased GFAP in both DRG and SC at chronic time points, highlights the similarities of the glial response in discogenic pain and neuropathy, and suggests that therapeutic strategies for chronic discogenic pain may be most effective by addressing pathologies in the spine as well as peripheral and central nervous systems.

## Figures and Tables

**Figure 1 ijms-25-01762-f001:**
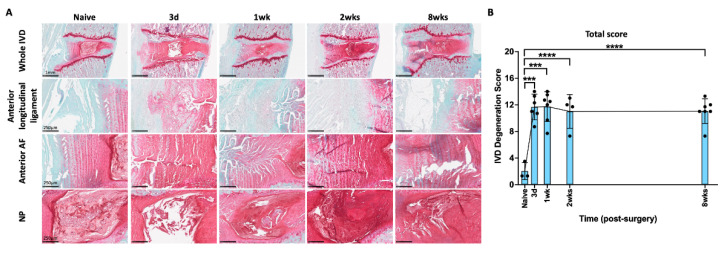
**AF puncture injury induced moderate to severe IVD degeneration.** (**A**) Representative images of IVD from different time points indicated that AF injury induced IVD degenerative changes with smaller and more fibrotic NP with fewer and/or clustered NP cells, less distinct NP-AF boundary, and disorganized AF with ruptured lamellae along the puncture track. Interestingly, GAG contents were observed within anterior longitudinal ligament at 3 days post injury, but the injured IVDs from other time points did not exhibit any observable GAG leakage. (**B**) The degeneration scores of the injured IVDs from 3 days, 1 week, 2 week, and 8 week time points were significantly higher than those from the naïve group. *** *p* < 0.001. **** *p* < 0.0001. The connecting line is drawn to visualize mean changes in the measurements with time. Each bar involves a separate experimental group sacrificed at that time point, and each dot represents a single animal.

**Figure 2 ijms-25-01762-f002:**
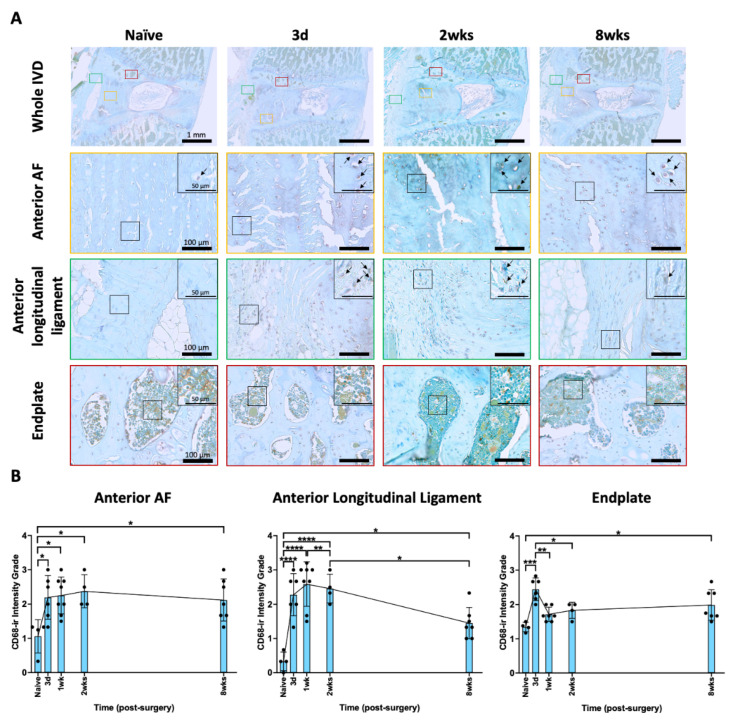
**AF puncture injury increased the level of CD68 in the spine.** (**A**) Representative images of CD68 IHC staining with toluidine blue in the entire IVD, anterior AF (yellow box), anterior longitudinal ligament (green box), and endplate (red box) at 3 days and 8 weeks after AF injury. The highlighted regions (black box) in anterior AF, anterior longitudinal ligament, and endplate show a closer look for positive, brown CD68 staining (indicated with arrow) vs. negative blue staining in each group from naïve to 8 weeks. (**B**) Semi-quantitative analysis showed that the CD68-ir in anterior AF increased at 3 days post-injury and remained elevated until 8 weeks post-injury, while the CD68-ir in the anterior longitudinal ligament and endplate were temporary, with peaks at 1 week and 3 days, respectively, and then significantly decreased. * *p* < 0.05. ** *p* < 0.01. *** *p* < 0.001. **** *p* < 0.0001. Naïve (n = 4), 3 days (n = 6), 1 week (n = 7), 2 weeks (n = 4), 8 weeks (n = 7). The connecting line is drawn to visualize mean changes in the measurements with time, and we note that each bar involves a separate experimental group sacrificed at that time point, and that each dot represents a single animal.

**Figure 3 ijms-25-01762-f003:**
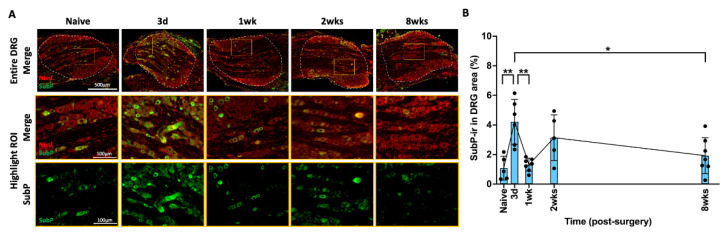
**AF injury increased SubP in DRG, with a peak at 3 days, and then restored to naïve level.** (**A**) Representative images of SubP IHC staining with Nissl stain in the DRG at different time points. The dotted circle is the region of interest (ROI) for quantifying the percentage of SubP-ir in each sample. The highlighted region (orange box) in DRG show a closer look for positive SubP staining with and without Nissl staining. (**B**) SubP-ir significantly increased and peaked at 3 days after injury, and then significantly decreased at 1 week and maintained at this level until the chronic timepoint of 8 weeks. * *p* < 0.05. ** *p* < 0.01. Naïve (n = 5), 3 days (n = 6), 1 week (n = 7), 2 weeks (n = 5), 8 weeks (n = 7). The connecting line is drawn to visualize mean changes in the measurements with time. Each bar involves a separate experimental group sacrificed at that time point, and each dot represents a single animal.

**Figure 4 ijms-25-01762-f004:**
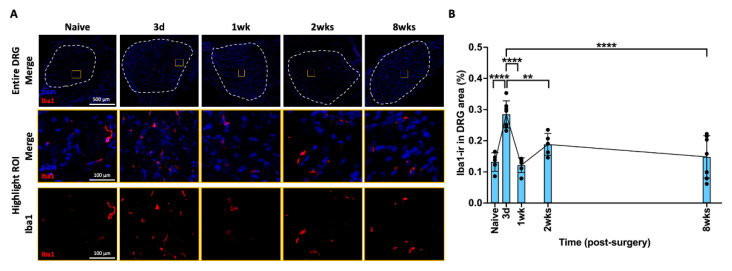
**AF injury increased Iba1 in DRG, with a peak at 3 days, and then restored to naïve level.** (**A**) Representative images of Iba1 IHC staining with DAPI in the DRG at different time points. The dotted circle is the region of interest (ROI) for quantifying the percentage of Iba1-ir in each sample. The highlighted region (orange box) in DRG show a closer look for positive Iba1 staining with and without DAPI staining. (**B**) Iba1-ir significantly increased and peaked at 3 days after injury, and then significantly decreased at 1 week and maintained at this level until the chronic timepoint of 8 weeks. ** *p* < 0.01. **** *p* < 0.0001. Naïve (n = 5), 3 days (n = 7), 1 week (n = 6), 2 weeks (n = 5), 8 weeks (n = 7). The connecting line is drawn to visualize mean changes in the measurements with time. Each bar involves a separate experimental group sacrificed at that time point, and each dot represents a single animal.

**Figure 5 ijms-25-01762-f005:**
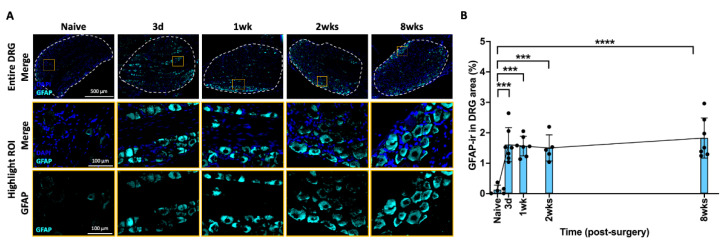
**AF injury increased GFAP in DRG from 3 days until 8 weeks.** (**A**) Representative images of GFAP IHC staining with DAPI in the DRG at different time points. The dotted circle is the region of interest (ROI) for quantifying the percentage of GFAP-ir in each sample. The highlighted region (orange box) in DRG show a closer look for positive GFAP staining with and without Nissl staining. (**B**) GFAP-ir significantly increased at 3 days and remained elevated until the chronic timepoint of 8 weeks. *** *p* < 0.001. **** *p* < 0.0001. Naïve (n = 5), 3 days (n = 7), 1 week (n = 7), 2 weeks (n = 5), 8 weeks (n = 7). The connecting line is drawn to visualize mean changes in the measurements with time. Each bar involves a separate experimental group sacrificed at that time point, and each dot represents a single animal.

**Figure 6 ijms-25-01762-f006:**
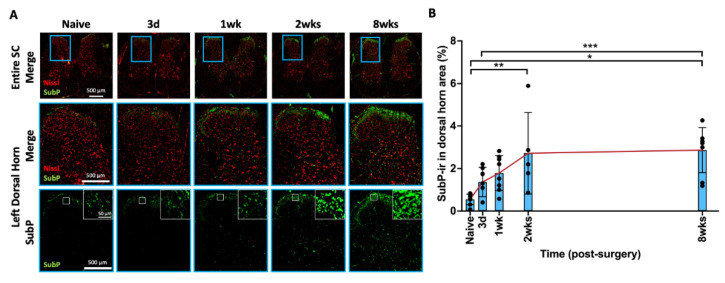
**AF injury increased SubP in SC, with a peak at 2 weeks, and then maintained until 8 weeks.** (**A**) Representative images of SubP IHC with Nissl stain in the entire spinal cord and left dorsal horn. The highlighted region (blue box) in SC shows a closer look for of left spinal cord dorsal horn for positive SubP staining with and without Nissl staining. The highlighted SubP images (white box) indicate density and staining intensity of SubP in the laminae of the SC dorsal horn. (**B**) AF injury significantly and continuously increased SubP-ir, which peaked at 2 weeks following injury and remained elevated until the timepoint of 8 weeks. * *p* < 0.05. ** *p* < 0.01. *** *p* < 0.001. Naïve (n = 8), 3 days (n = 8), 1 week (n = 8), 2 weeks (n = 5), and 8 weeks (n = 8). The connecting line is drawn to visualize mean changes in the measurements with time. Each bar involves a separate experimental group sacrificed at that time point, and each dot represents a single animal.

**Figure 7 ijms-25-01762-f007:**
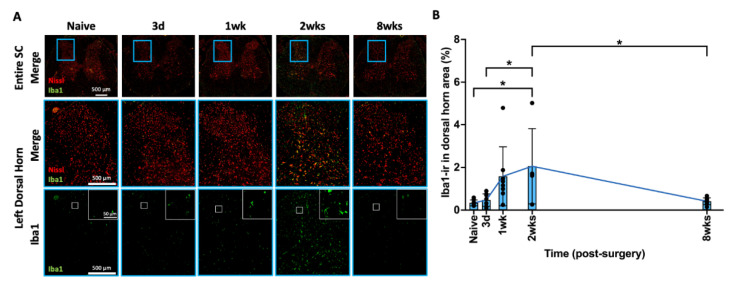
**AF injury increased Iba in SC, with a peak at 2 weeks, and then declined at 8 weeks.** (**A**) Representative images of Iba1 IHC staining with Nissl stain in the entire spinal cord and left dorsal horn. The highlighted region (blue box) in SC shows a closer look for of left spinal cord dorsal horn for positive Iba1 staining with and without Nissl staining. The highlighted Iba1 images (white box) indicate density and staining intensity of Iba1 in the SC dorsal horn. (**B**) AF injury significantly and continuously increased Iba1-ir and peaked at 2 weeks following injury and declined until the chronic timepoint of 8 weeks. * *p* < 0.05. Naïve (n = 8), 3 days (n = 8), 1 week (n = 8), 2 weeks (n = 5), and 8 weeks (n = 8). The connecting line is drawn to visualize mean changes in the measurements with time. Each bar involves a separate experimental group sacrificed at that time point, and each dot represents a single animal.

**Figure 8 ijms-25-01762-f008:**
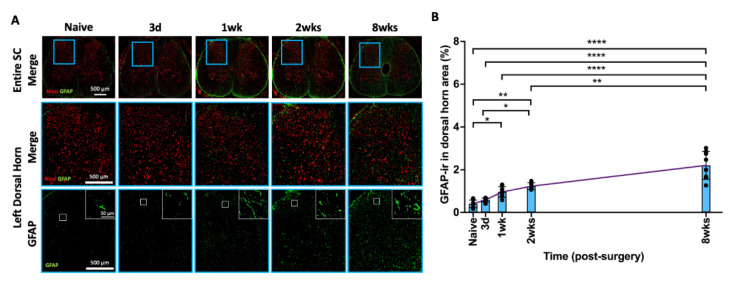
**AF injury continuously and significantly increased GFAP in SC from 3 days until 8 weeks.** (**A**) Representative images of GFAP IHC with Nissl stain in the entire spinal cord and left dorsal horn. The highlighted region (blue box) in SC shows a closer look for of left spinal cord dorsal horn for positive GFAP staining with and without Nissl staining. The highlighted GFAP images (white box) indicate morphology of astrocytes as well as density and staining intensity of GFAP in the SC dorsal horn. (**B**) AF injury significantly and gradually increased GFAP-ir until the chronic timepoint of 8 weeks. * *p* < 0.05. ** *p* < 0.01. **** *p* < 0.0001. Naïve (n = 8), 3 days (n = 8), 1 week (n = 8), 2 weeks (n = 5), and 8 weeks (n = 8). The connecting line is drawn to visualize mean changes in the measurements with time. Each bar involves a separate experimental group sacrificed at that time point, and each dot represents a single animal.

**Figure 9 ijms-25-01762-f009:**
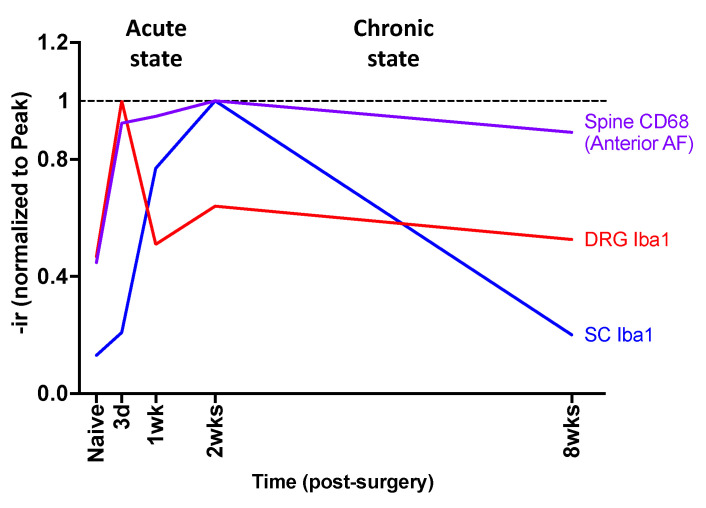
Temporal crosstalk of inflammatory responses between spine, DRG, and SC at acute and chronic states following AF injury. AF injury induced increased spine CD68-ir (anterior AF, purple line) and DRG Iba1-ir (red line) with peak at 3 days post-injury, followed by SC Iba1-ir (blue line) with a peak at 2 weeks post-injury.

**Figure 10 ijms-25-01762-f010:**
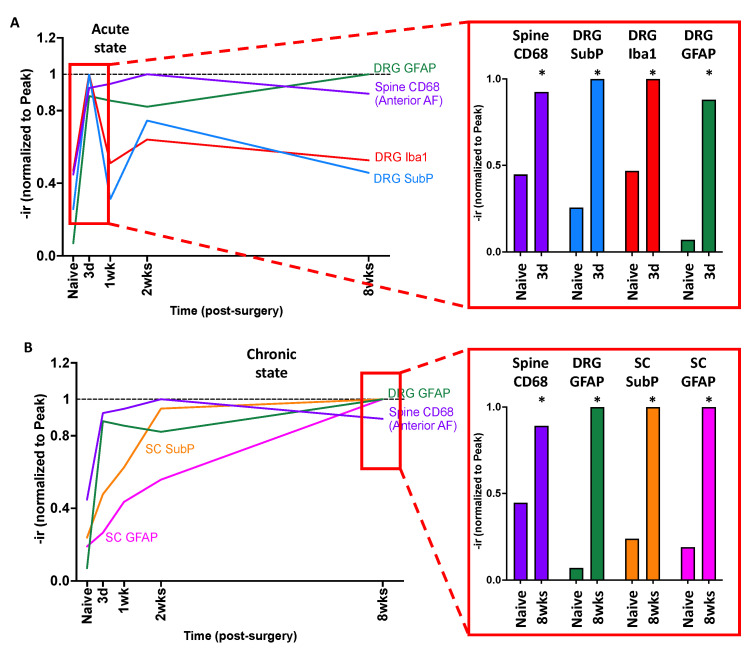
Temporal crosstalk of inflammatory responses and sensitization between spine, DRG, and SC in acute and chronic states following AF injury. (**A**) AF injury induced acute responses in DRG with increased SubP-, Iba1-, and GFAP-ir at 3 days post-injury. Both DRG SubP- and Iba1-ir restored at 1 week post-injury, but DRG GFAP remained at the elevated level. (**B**) AF injury induced chronic changes in DRG GFAP-ir, as well as SC SubP- and GFAP-ir, with a peak at 8 weeks post-injury. * *p* < 0.05 compared to naïve rats.

**Figure 11 ijms-25-01762-f011:**
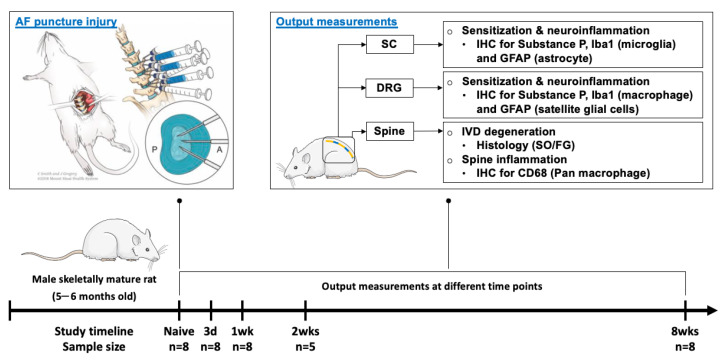
**Schematic image of AF puncture injury with anterior approach surgery.** Surgical methods, study design, output measurements, and timeline. The rats that received AF injury were euthanized at different time points, i.e., 3 days, 1 week, 2 weeks, and 8 weeks after injury, while the naïve rats did not receive any surgical interventions and were used as a control group. After euthanization, the rat lumbar spines, L2 DRGs, and lumbar SCs were isolated for histological and immunohistochemical analyses. “A” and “P” represents the anterior and posterior region of the IVD.

## Data Availability

The data that support the findings of this study are available from the corresponding author, James C. Iatridis, upon request.

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
