# Peer review of "Annulus Fibrosus Injury Induces Acute Neuroinflammation and Chronic Glial Response in Dorsal Root Ganglion and Spinal Cord—An In Vivo Rat Discogenic Pain Model"

_ijms, 2024, doi:10.3390/ijms25031762_

Round 1
Reviewer 1 Report
Comments and Suggestions for Authors
This is a very comprehensive, well-designed study to characterize the time course of neuroinflammatory changes in the peripheral and central nervous systems. The authors have done a commendable job and I have no critiques to provide.
Author Response
Thanks for your supportive comments and careful read.
Reviewer 2 Report
Comments and Suggestions for Authors
The paper deals with the important problem of acute and chronic changes after annulus fibrosus injury. The sequence of development of neuroinflammation and its extent is particularly important because it has translational significance for clinical practice.
The following shortcomings were observed:
1. The title says '...induces acute neuroinflammation and chronic astroglial response in dorsal root ganglion and spinal cord...' - from which it follows that satellite cells are a type of astroglia. However, astroglia originate from the neural tube, and satellite cells from the neural crest. Both types of cells are glia, but only cells of the central nervous system are astroglia. The title would be more correct if it said '...chronic glial response...' The same error of identifying satellite cells with astroglia is repeated in the abstract and in the main text and should be corrected.
2. In line 133 you say '...GFAP as a marker of neuronal remodeling...but GFAP is an intermediate filament that primarily indicates remodeling and functional changes of glial cells. Can you better explain the role of GFAP in glial cells and how it contributes to the functional changes of glial cells.
3. From the design of the study, only 3 animals served as controls. It is not clear when they were sacrificed. It is also not clear what number of animals was used for which of the mentioned methods, and this is important in order to be able to assess the validity of the statistics.
4. Explain how you did the quantification of individual staining - how many animals you had by each method, how many sections or how many ROIs per slide.
5. In Figures 2-9, column B is a bar chart. The bars are connected to each other by lines, but this is not correct because these are independent experiments. It is not about one and the same animal that was followed through time, but about groups of animals that were sacrificed at different times.
6. In some images, the magnification is not sufficient to be able to assess the characteristic immunostaining, for example, for GFAP, Iba and SubP. Please provide inserts with higher magnifications where there are only a few immunopositive cells.
7. You explained why you didn't do the behavior, but still explain how you checked for changes in the sensation of pain or the appearance of allodynia in order to be sure that the injury was done correctly. Did the animals lose weight or did they show any changes in their behavior?
8. Please indicate which decalcification procedure you used.
Author Response
- The title says '...induces acute neuroinflammation and chronic astroglial response in dorsal root ganglion and spinal cord...' - from which it follows that satellite cells are a type of astroglia. However, astroglia originate from the neural tube, and satellite cells from the neural crest. Both types of cells are glia, but only cells of the central nervous system are astroglia. The title would be more correct if it said '...chronic glial response...' The same error of identifying satellite cells with astroglia is repeated in the abstract and in the main text and should be corrected.
- Response: Thanks for the suggestion. We agree that glial is a better terminology to describe both satellite glial cells and astrocytes, and we have modified the title and abstract as suggested.
- In line 133 you say '...GFAP as a marker of neuronal remodeling...but GFAP is an intermediate filament that primarily indicates remodeling and functional changes of glial cells. Can you better explain the role of GFAP in glial cells and how it contributes to the functional changes of glial cells.
- Response: It is a great suggestion. As highlighted by Reviewer #2, GFAP is a key intermediate filament protein present in various types of glial cells, including astrocytes in the central nervous system and satellite glial cells in the peripheral nervous system. GFAP makes up the cytoskeletal framework to maintain the structural integrity of glial cells, and regulates concentrations of neurotransmitters and ions for optimal neuronal function during normal physiological conditions. Importantly, in response to nerve injury and inflammatory pain, GFAP demonstrates upregulation in both satellite glial cells (SGC) and astrocytes, indicating that both SGC and astrocytes are activated. The activated glial cells, particularly astrocytes, undergo changes in morphology characterized by the proliferation and alteration of processes, becoming thicker and longer. Moreover, the activation of glial cells, as evidenced by increased GFAP expression, is associated with functional changes that contribute to pain. These activated glial cells release various signaling molecules, including cytokines and chemokines, along with increased production of potassium and glutamate. This cascade of events sensitizes neurons, amplifying their responsiveness and contributing to the manifestation of pain. Text has been added into the Introduction section (lines 123-133) to emphasize the role of GFAP and how it contributes to functional changes of glial cells.
- From the design of the study, only 3 animals served as controls. It is not clear when they were sacrificed. It is also not clear what number of animals was used for which of the mentioned methods, and this is important in order to be able to assess the validity of the statistics.
- Response: We would like to clarify that our Naive control group comprised 5 animals that underwent no surgical procedures and were sacrificed at 5-6 months old.
- Regarding the sample size breakdown, the Naive group consisted of 5 animals, while the AF injury group encompassed a total of 32 rats subjected to annular puncture surgery. These rats were euthanized at different time points post-injury: 3 days (n=8), 1 week (n=8), 2 weeks (n=5), and 8 weeks (n=8), as illustrated in Figure 1. This information has been incorporated into the Study design of Materials and Methods section 4.1 to ensure clarity (lines 779-780). We further clarify that the original manuscript (lines 772-782) contained limitations to explain our control choice and clarifies why we believe this study is sufficiently controlled.
- Explain how you did the quantification of individual staining - how many animals you had by each method, how many sections or how many ROIs per slide.
- Response: The quantification of individual staining in the DRG involved a precise selection process based on orientation and section level. DRG samples towards the center were chosen for enhanced neuronal density, and axial/ventral sections for equal soma and axon distribution (Sperry+ J Comp Neurol 2020). Exclusion criteria were applied to eliminate sections with primary neuronal areas cut off or containing autofluorescence. Nissl and DAPI counterstains, intensified for better visualization, aided in outlining the region of interest (ROI) containing all neuronal bodies. Using a standardized immunoreactivity (ir) threshold, positive pixels for SubP-ir, Iba1-ir, and GFAP-ir were calculated relative to the ROI area. The analysis incorporated both right and left DRGs, with averaged percentage areas per section. This data was then used to represent values per animal per time point. This information has been incorporated into the Materials and Methods section 4.6 (line xxx). The sample sizes slightly varied due to excluded samples. For SubP: Naive (n=5), 3d (n=6), 1wk (n=7), 2wk (n=5), 8wk (n=7). For Iba1: Naive (n=5), 3d (n=7), 1wk (n=6), 2wk (n=5), 8wk (n=7). For GFAP: Naive (n=5), 3d (n=7), 1wk (n=7), 2wk (n=5), 8wk (n=7). This information has been incorporated into the Materials and Methods section 4.6 (lines 928-940) as well as into the figure legend of individual staining.
- For the spinal cord, two sections with two ROI’s for each section (left and right dorsal horn) were examined, analyzed, and averaged for each animal for each staining. Specifically, left and right spinal cord dorsal horns (lamina I-V, primarily for sensory input reception) for each section were outlined based on guidance from the Nissl stain TXR channel and treated as separate ROIs. Each spinal cord ROI was analyzed using a standardized immunoreactivity (ir) threshold to calculate positive pixels of SubP-ir, Iba1-ir and GFAP-ir relative to the ROI area. The percentage areas of immunopositive staining from the right and left dorsal horn ROIs were averaged per section, and the averaged values from the two sections were then used to represent a single value per animal per time point for statistical analysis. This detailed procedure was clarified and added to the Materials and Methods Section 4.7. The sample size for spinal cord analysis comprised Naive (n = 8), 3d (n = 8), 1wk (n = 8), 2wk (n = 5), and 8wk (n = 8). This information was added to the Materials and Methods Section 4.7 (lines 979-990) as well as the figure legend of individual staining.
- In Figures 2-9, column B is a bar chart. The bars are connected to each other by lines, but this is not correct because these are independent experiments. It is not about one and the same animal that was followed through time, but about groups of animals that were sacrificed at different times.
- Response: Thanks for your suggestion. We would like to clarify that these connecting lines are intended to visually facilitate the changes with time of each measurement, and highlight that experimental groups involved animals sacrificed at distinct time points (represented by dots in the bar graphs). To insure that the reader fully understands we are not tracking individual animals with time, each figure legend now includes the statement: ‘The connecting line is drawn to visualize mean changes in the measurements with time, and we note that each bar involves a separate experimental group sacrificed at that time point, and that each dot represents a single animal.’
- In some images, the magnification is not sufficient to be able to assess the characteristic immunostaining, for example, for GFAP, Iba and SubP. Please provide inserts with higher magnifications where there are only a few immunopositive cells.
- Response: Thanks for the comments. We have included additional images with higher magnification specifically focusing on areas with a limited number of immunopositive cells. This modification aims to ensure a more detailed and clearer representation of the characteristic immunostaining for CD68 in spine, as well as GFAP, Iba, and SubP in spinal cord dorsal horn (Figures 2, 6-8). We believe that these additions will enhance the overall interpretability and utility of the images.
- You explained why you didn't do the behavior, but still explain how you checked for changes in the sensation of pain or the appearance of allodynia in order to be sure that the injury was done correctly. Did the animals lose weight or did they show any changes in their behavior?
- Response: We did not measure the body weight of the rats in this study, to avoid additional handling and stress for the animal, since our previous studies with this injury model [Lai et al. J Orthop Res, 2015; Lai et al. Spine J, 2016; Evashwick-Rogler et al. JOR Spine, 2018; Mosley et al. Spine (Phila Pa 1976), 2019], consistently demonstrated a continuous increase in body weight with no significant differences between groups at any time points post-injury. Additionally, although pain-related behaviors were not directly assessed in this study, we took specific measures to ensure the accurate execution of the injury. After surgery, we closely monitored the animals general cage activities, including mobility, staggering, interaction with the researcher, and body conditioning score. Animals with spinal injuries did not present discernible differences compared to Naive. Furthermore, there were no intraoperative complications for this cohort. These findings strongly suggested that reported measurements are related to the spinal injury and that the AF injury procedures were well tolerated by the rats and did not significantly affect their general health. Specific details about weight measurement and general cage activities have been added into the Results section 2.1 (lines 153-254).
- Please indicate which decalcification procedure you used.
- Response: The spine specimens were decalcified in formic acid over 3 days with 3 changes, and this information has been added into the Materials and Methods section 4.4 (lines 853-854).

Reviewer 3 Report
Comments and Suggestions for Authors
General comments:
- This is a very interesting study, it is about an updated theme, with the possibility to be translated for companion animals. The material and methods and results section is confusing and, regarding, structure, the manuscript has to have an improvement. In the conclusion and discussion, the authors should explore how this will help in the therapeutic approach.
Specific comments:
Introduction
- Lines 45-50, 64-67, 98-10, 119-122: Some bibliographic references is missing in these paragraphs.
- Line 127-130: I suggest to move this paragraph to the material and methods section;
- line 130-135: The authors should describe the aim of the study and not I they perform it. Also, the aim should be rewritten shortly and directly to be clearer and simpler for the readers. The hypothesis of the study is direct and clear.
Material and methods/ Results:
- In my opinion point 2 (2.1) should not be in the results section. It is the description of the surgical methods and the study design output measurements and timeline. This should be in the material and methods section.
- In point 2.2, methods are developed, followed by the study design and then results. It is confusing...It should be done seprately.
- In general, I suggest to the authors to make it clearer the separation between material and methods and results throughout the study.
Discussion
- It is poor regarding the recent bibliography. The authors should explore and compare with recent references.
Conclusion
- I suggest that the authors re-write the conclusion more clearly and directly. It may help to split into paragraphs.
Author Response
- Please indicate the rats gender in "study design".
- Response: Male rats were used in this study, and the sex has been added in the Study design of the Materials and Methods section 4.1 (line 805).
- line 593: what is "ir"?
- Response: “ir” denotes immunoreactivity, with its initial definition provided in Results section 2.3 of the manuscript. To enhance clarity and understanding, this definition has been included once again in the Materials and Methods section 4.5 (line 878).
- All figures presented must be readable, therefore I strongly recommend authors to improve improve all figures. I.e., IHC, IF labels, X-y-axis labels, p value labels, data points etc. Without these changes, its extremely hard to read figures.
- Response: Thank you for your recommendations. We have increased image and text sizes to enhance the readability of all figures. We trust that these adjustments will contribute to a clearer and more accessible presentation of the figures in the manuscript. We welcome any additional and more specific suggested changes to the figures.
- This manuscript is nicely written and provides insightful evaluation of the model. It can be accepted after improving figures.
- Response: Thanks for your supportive comments. All figures have been modified as suggested, and all comments have been addressed.
Reviewer 4 Report
Comments and Suggestions for Authors
This study by Lai A et al., evaluated the inflammatory pattern in the rat model of chronic painful invertebrate degeneration. By systematic temporal analysis, authors concluded that AF induced acute spinal, DRG, and SC inflammation.
My concerns:
1) Please indicate the rats gender in "study design".
2) line 593: what is "ir"?
All figures presented must be readable, therefore I strongly recommend authors to improve improve all figures. I.e., IHC, IF labels, X-y-axis labels, p value labels, data points etc. Without these changes, its extremely hard to read figures.
This manuscript is nicely written and provides insightful evaluation of the model. It can be accepted after improving figures.
Author Response
Introduction
- Lines 45-50, 64-67, 98-10, 119-122: Some bibliographic references is missing in these paragraphs.
- Response: Thank you for bringing this to our attention. We have carefully reviewed the specified paragraphs, and added the following references to support our statements:
- For the ineffectiveness of surgical interventions for IVDD: Anderson+ Spine (Phila Pa 1976) 2015; Endler+ ​​Bone Joint J 2019; Schizas+ Eur Cell Mater 2010
- To provide evidence for SubP, microglia, and astrocyte involvement in neuropathic pain pathogenesis: Abbadie+ Neuroscience 1996; Lee+ Neurosci Res 2007; Matsuda+ J Anesth 2019; Moss+ Pain 2007; Myers+ Drug Discov Today 2006; Wang+ PLoS One 2009
- Emphasizing studies that have been previously done to assess pain-related behaviors in IVDD: Lai+ J Orthop Res 2015; Lai+ Spine J 2016; Mosley+ Ann N Y Acad Sci 2017; Wawrose+ JOR Spine 2022
- We believe these additions enhance the rigor and broader context of this study manuscript.
- Line 127-130: I suggest to move this paragraph to the material and methods section;
- Response: Thanks for the suggestion, and we moved this statement to the Study design of the Materials and Methods section 4.1 (lines 808-810).
- line 130-135: The authors should describe the aim of the study and not I they perform it. Also, the aim should be rewritten shortly and directly to be clearer and simpler for the readers. The hypothesis of the study is direct and clear.
- Response: Thank you for the suggestion. We have revised the aims of this study to enhance clarity and directness (lines 137-146).
Material and methods/ Results:
- In my opinion point 2 (2.1) should not be in the results section. It is the description of the surgical methods and the study design output measurements and timeline. This should be in the material and methods section.
- Response: Thanks for the suggestion, and the description of the surgical methods as well as the study design output measurements and timeline were removed from the Results section 2.1.
- In point 2.2, methods are developed, followed by the study design and then results. It is confusing...It should be done seprately.
- Response: That’s a good suggestion. We have revised the manuscript to make a clearer separation between the Materials and Methods section and the Results section.
- In general, I suggest to the authors to make it clearer the separation between material and methods and results throughout the study.
- Response: Thanks again for the suggestion. As above, we have revised the manuscript to make a clearer separation between the Materials and Methods section and the Results section.
Discussion
- It is poor regarding the recent bibliography. The authors should explore and compare with recent references.
- Response: Thanks for your feedback regarding the need for a more robust recent bibliography. In response to this suggestion, we conducted a thorough search on Pubmed and have incorporated two additional references (Sun et al., J Neuroimmune Pharmacol, 2022; Zhu et al., Oxid Med Cell Longev, 2022) into the Discussion section (line xxx). These studies, demonstrating an early microglial responses following spared nerve injury as well as increased astrocytes in late-phase diabetic neuropathic pain, align closely with our findings
- We believe these recent additions significantly strengthen the manuscript. If you have further recommendations or specific areas where recent references would be beneficial, we welcome your guidance.
Conclusion
- I suggest that the authors re-write the conclusion more clearly and directly. It may help to split into paragraphs.
- Response: Thanks for the suggestion. The conclusion has been revised to enhance clarity and directness, and we have incorporated paragraph breaks to better organize the content. We believe these changes will contribute to a more effective and accessible conclusion.
Round 2
Reviewer 2 Report
Comments and Suggestions for Authors
The authors have accepted the remarks, and the entire text has also been significantly improved. I believe that this is an important paper for this area of research.
Reviewer 3 Report
Comments and Suggestions for Authors
Dear authors thank you for your revisions.
I think the authors coped with all the suggestions made and responded to all questions.
In my opinion, after this revision, the manuscript is acceptable for publication.